# A supertough electro-tendon based on spider silk composites

Liang Pan[1], Fan Wang[2], Yuan Cheng [3], Wan Ru Leow[1], Yong-Wei Zhang[3], Ming Wang[1], Pingqiang Cai [1], Baohua Ji[4], Dechang Li [4✉] & Xiaodong Chen [1✉]

Compared to transmission systems based on shafts and gears, tendon-driven systems offer a simpler and more dexterous way to transmit actuation force in robotic hands. However, current tendon fibers have low toughness and suffer from large friction, limiting the further development of tendon-driven robotic hands. Here, we report a super tough electro-tendon based on spider silk which has a toughness of 420 MJ/m$^3$ and conductivity of 1,077 S/cm. The electro-tendon, mechanically toughened by single-wall carbon nanotubes (SWCNTs) and electrically enhanced by PEDOT:PSS, can withstand more than 40,000 bending-stretching cycles without changes in conductivity. Because the electro-tendon can simultaneously transmit signals and force from the sensing and actuating systems, we use it to replace the single functional tendon in humanoid robotic hand to perform grasping functions without additional wiring and circuit components. This material is expected to pave the way for the development of robots and various applications in advanced manufacturing and engineering.

[1] Innovative Centre for Flexible Devices (iFLEX), School of Materials Science and Engineering, Nanyang Technological University, 50 Nanyang Avenue, Singapore 639798, Singapore. [2] Biomechanics and Biomaterials Laboratory, Department of Applied Mechanics, Beijing Institute of Technology, Beijing 100081, China. [3] Institute of High Performance Computing, Agency for Science Technology and Research (A*STAR), 1 Fusionopolis Way, Singapore 138632, Singapore. [4] Institute of Applied Mechanics, Department of Engineering Mechanics, Zhejiang University, Hangzhou 310027, China. ✉email: dcli@zju.edu.cn; chenxd@ntu.edu.sg

The loss of appendages can severely affect a person's quality of life. As a result, humanoid robotic hands with capabilities comparable to human limbs have been actively explored for use as prosthesis[1–3]. The core component of these robotic hands is the tendon-driven transmission system, which relies on a fiber that resembles the human tendon to transmit power from actuators to joints. Compared to other systems based on leverages, shafts or gears, tendon-driven transmission is simpler in design and offers better dexterity and flexibility. It has been widely used in humanoid robotic hands, such as the Okada Hand[4], Utah/MIT Hand[5], and DLR Hand[6]. However, current tendon fibers, which are typically made from nylon, silicone rubber, or polyethylene terephthalate (PET), have low toughness and therefore, cannot endure many bending and stretching cycles. These fibers also suffer from large friction along the narrow tendon path, further lowering their durability[5,7,8]. Given many of these tendons are non-conductive and have a single function, integrating wires for the transmission of electrical signals from sensing systems and additional fibers as tendons onto a slender robotic finger with the size of a human hand is challenging[2,3,9].

Currently, there are no materials or systems that simultaneously have high toughness, conductivity, and stretchability for

mechanical engineering applications such as tendons in robotic hands[10,11]. Polymer-based conductors typically show low toughness (<100 MJ/m³) and poor conductivity (<100 S/cm)[12–17]. For example, the toughness of PDMS-based conductors is only around 0.6–10 MJ/m³[18–23]. Although traditional metals such as Au, Al, and Cu have excellent conductivity, they have low toughness (around 1–10 MJ/m³)[24] (Fig. 1a and Supplementary Table 1), making these materials unsuitable for robotic applications. There is, therefore, a demand for materials that are simultaneously flexible, conductive and durable, so they can be easily integrated on a finger to achieve human-like performance and functionality.

Here, we report an "electro-tendon" based on spider silk that has a toughness of 420 MJ/m³ and conductivity of 1077 S/cm (pink star in Fig. 1a), properties that are better than current flexible and stretchable conductors. The electro-tendon makes from the *Nephila pilipes* spider dragline silk, single-walled carbon nanotube (SWCNT) and poly(3,4-ethylenedioxythiophene) polystyrene sulfonate (PEDOT:PSS). We show this electro-tendon can bend and stretch more than 40,000 cycles without any change in conductivity. When attaching to a pressure sensor and mounting on a 3D-printed human-like robotic finger, the electro-

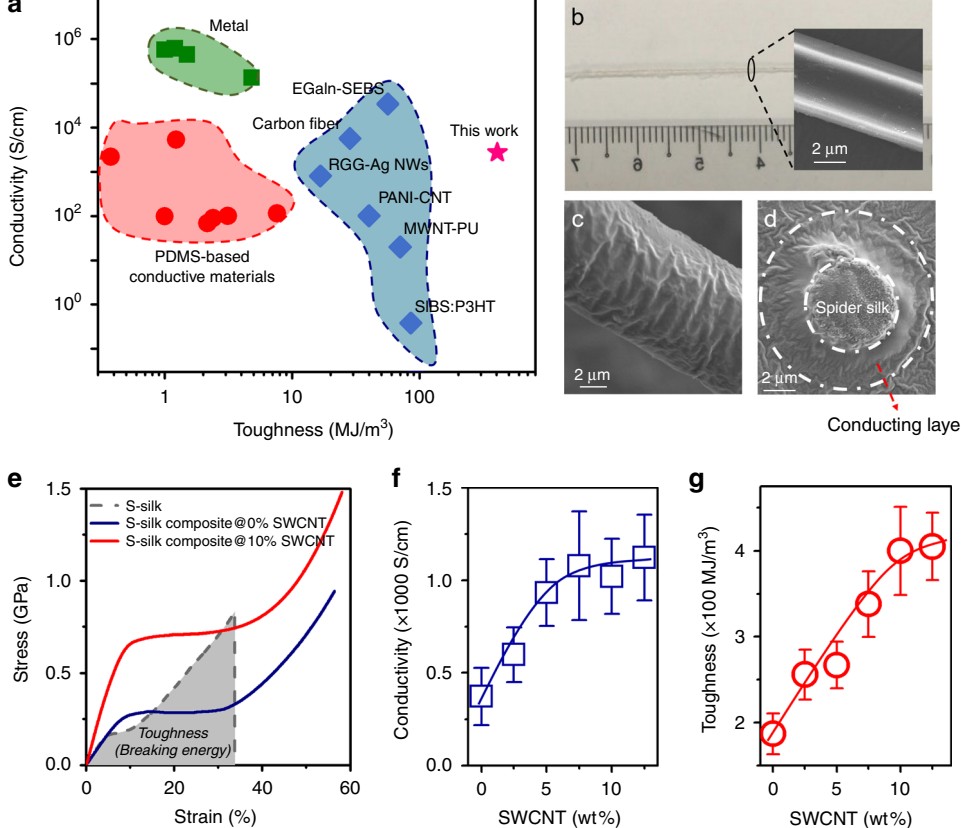

**Fig. 1 Toughness and conductivity of spider silk composites. a** Graph shows the conductivity and toughness of different flexible materials. Green represents metals, red is PDMS-based stretchable conductors, and blue represents other special conducting materials/structures. Pink star represents S-silk@10% SWCNT composites described in this work. **b** Optical image of a bundle of raw dragline silk from *Nephila pilipes*. Inset shows a single silk fiber has a very smooth surface. **c** SEM image showing the wrinkled surface of a single fiber of S-silk@10% SWCNT composite formed from the intrinsic shrinkage of the spider silk after immersion in water during PEDOT:PSS and SWCNT coating. The wrinkled structure prevented any changes in the conductive path, allowing the S-silk composite to maintain its conductivity during stretching and compression. **d** Cross-sectional scanning electron microscopic image of a spider silk composite. The core is spider silk, and the diameter is about 3–4 μm. The outer conducting layer is about 2 μm. **e** Stress–strain curves of natural spider silk (S-silk), spider silk with PEDOT:PSS@0%SWCNT (S-silk composite@0%SWCNT), spider silk with PEDOT:PSS@10%SWCNT (S-silk composite@10%SWCNT). Area under gray dotted curve represents toughness of S-silk. Toughness is defined as the energy needed to break the silk. **f**, **g** Conductivity and toughness of the S-silk composite increased with increasing weight percent of SWCNT, before experiencing saturation at 12.5 wt%. The maximum conductivity and toughness achieved were 1077 S/cm and 420 MJ/m³, respectively. Our S-silk composite is more conductive and tougher than the other flexible materials shown in Fig. 1a. The error bars in **f** and **g** show standard deviations based on 50 independent samples.

tendon enables the robotic fingers to respond and capture various objects without damaging the objects. This feat is because of the stable transfer of both electrical signals through the tendon fiber from the pressure sensor and force signals from the actuating system. Because the electro-tendon can transmit signals to and from both the actuating and sensing systems, it can be mounted on a slender robotic finger without the need for additional wires or circuit components, significantly simplifying any robotic setup.

## Results

**Spider silk composites design and characterization**. Spider silk (S-silk), hailed as a "super fiber" (Supplementary Fig. 1 and Supplementary Movie 1), is one of the toughest natural materials in the world, outperforming the best synthetic high-performance fibers available today[25–30]. For instance, the toughness of *Nephila pilipes* spider dragline silk is ~160 MJ/m³, while that of Kevlar (Dupont Advanced Fiber Systems), the material used in bullet-proof body armor, is ~50 MJ/m³ [31]. Given its toughness, S-silk is an attractive candidate for fabricating electro-tendons for use in humanoid robotic hands. To make S-silk with high conductivity, a layer of PEDOT:PSS was coated, due to their high mechanical flexibility, good dispersibility, and excellent electrical conductivity with 3000 S/cm after annealing[32–34]. Because of hydrophilic PSS, PEDOT:PSS has good adhesion with the processed S-silk to form a conformal conducting layer (Fig. 1b, c and Supplementary Fig. 2 for detailed fabrication procedure)[32]. On the other hands, we have introduced SWCNT into the silk to improve the toughness. SWCNT is a fascinating material that is a one atom thick layer of graphite rolled into a cylinder with a 1 nm diameter. It has exceptional mechanical modulus of 1 TPa and tensile strength of 100 GPa. Thus, SWCNT are often used for mechanical enhancement in many material systems[34–38]. More importantly, SWCNT in this work can further improve the conductivity of S-silk through electronic density transfer from PEDOT to SWCNT[39,40]. Supplementary Fig. 3 shows the optical image of modified spider silk. As compared to raw spider silk, the color of the S-silk composite changed into black. Then, we confirmed the presence of SWCNT in the silk through G band in the Raman spectrum of the sample's cross-section (Fig. 1d, Supplementary Fig. 4). The outer layer of conducitng layer is epoxy resin which was used to bury the spider silk for good corss-section SEM images. As the weight percent of SWCNT increased (0–12.5 wt%), the intensity of the G band increased but the stress–strain curve remained nearly unchanged until 12.5 wt% due to poor dispersion of SWCNT in water (Supplementary Fig. 5). Because there were no differences in mechanical properties between the composite with 12.5 and 10 wt% SWCNT, we used silk composites containing 10 wt% SWCNT for the robotic finger experiments.

The conductivity of the S-silk composites enhanced with increasing SWCNT content, achieving 1077 S/cm with 7.5 wt% SWCNT (Fig. 1f). Moreover, the toughness of S-silk (determined from the area under the strain–stress curve) increased 2–3 times to 420 MJ/m³ upon addition of 10 wt% SWCNT (Fig. 1g). Other mechanical parameters such as Young's modulus and strength were also improved with increasing SWCNT content. And, the maximum strain (~60%) was nearly unchanged along with the content of SWCNT (Supplementary Fig. 6). To enable stable electrical signal transmission under strain, we made the conducting layer wrinkled by exploiting the intrinsic shrinkage of S-silk in aqueous solution (Fig. 1d and Supplementary Fig. 2ii)[41,42]. The wrinkled structure, which flattened upon stretching, prevented any changes in the conductive path, allowing the conductivity of the S-silk composites to remain nearly unchanged even after >36,5000 cycles of stretching and compression between 0% and 15% strain

(the maximum strain of the human tendon is ~15%) (Supplementary Figs. 7 and 8). From Supplementary Fig. 5b, when applied larger strain ~60%, the conductivity changed about 5%. Thus, before rupture, the S-silk composite can stably transmit the electrical signals.

**Dissipative particle dynamic (DPD) simulation**. To understand how SWCNT improve the mechanical properties of S-silk at the microscopic scale, DPD simulation was performed. Here we adopted a coarse-grained description[43,44] of silk proteins and SWCNT to investigate the structural evolution of amorphous ($3_1$-helix and β-turn) and crystalline (β-sheet nanocrystal) structures of silk in the presence of SWCNT under different strains. Every 9 water molecules are represented by one hydrophilic "w" bead, as shown in Supplementary Fig. 9a. Silk peptides are described as multiblock copolymer chains composed of hydrophobic "a" and hydrophilic "b" beads, with each bead representing three amino acids in the β-sheet crystalline and amorphous domains, respectively, as shown in Supplementary Fig. 9b. The CG single silk peptide is represented by the sequence $[b]_{16}-[a]_5-[b]_7-[a]_5-[b]_7$ in this study. The CG beads are connected by harmonic potentials[43]. To include SWCNT into the DPD simulations, each SWCNT is also described as CG beads connected by harmonic potential, as shown in Supplementary Fig. 9c.

Figure 2a, b show a snapshot of the coarse-grained S-silk composite. At the beginning of applying strain, the amorphous regions in natural S-silk gradually unfolded[28]. The hydrogen bond in the amorphous structure was broken with strain increasing. In the case of the S-silk composite, the interactions between SWCNT and silk proteins induced higher strength than that without SWCNT. With strain further increasing, the composite experiences higher normalized stress due to the bridge effect between SWCNT and the silk proteins (Fig. 2c–f, Supplementary Fig. 10a, b and movies 2 and 3). The S-silk composite fractured more difficult than S-silk at a lower critical strain, showing the composite has much tougher and better mechanical properties. In the simulation, we defined fracture as the number of bridges crossing any cross section to be <1. DPD simulation of toughness, Young's modulus, and strength as a function of SWCNT wt% agreed well with experimental data (Fig. 2g–i), suggesting the importance of SWCNT for improving the mechanical properties of spider silk.

**Endurability of spider silk composite-based tendon**. As proof-of-concept, we used the S-silk composite containing 10 wt% SWCNT as an electro-tendon to assemble a robotic hand that can perform basic functions of human hands. The robotic finger, made from polylactic acid (PLA), was 3D-printed to the same size of a human finger bone. The electro-tendon was attached to the inner side of the robotic finger and held in place using a silicone-based extensor (Fig. 3a, Supplementary Fig. 11). We measured the changes in the angle of the robotic finger relative to the vertical axis and the change in length of the tendon when the index finger experiences full bending. The initial angle of the index finger at resting state was about 8° (Fig. 3b). When the finger was bent to its final state pulled by the tendon from 0 to 5.2 cm, the angle changed to 73° (Fig. 3c). And, this whole process took ~1.5 s, according with the human bending process (Fig. 3d).

To confirm the durability of the S-silk composite-based tendon, we conducted cyclic bending tests and compared the results with other tendon materials (i.e. natural S-silk, nylon fiber, carbon fiber, steel fiber, and PDMS fiber) (Fig. 3e). Materials with higher toughness were clearly more durable as shown by the ability to withstand greater number of bending cycles. The finger

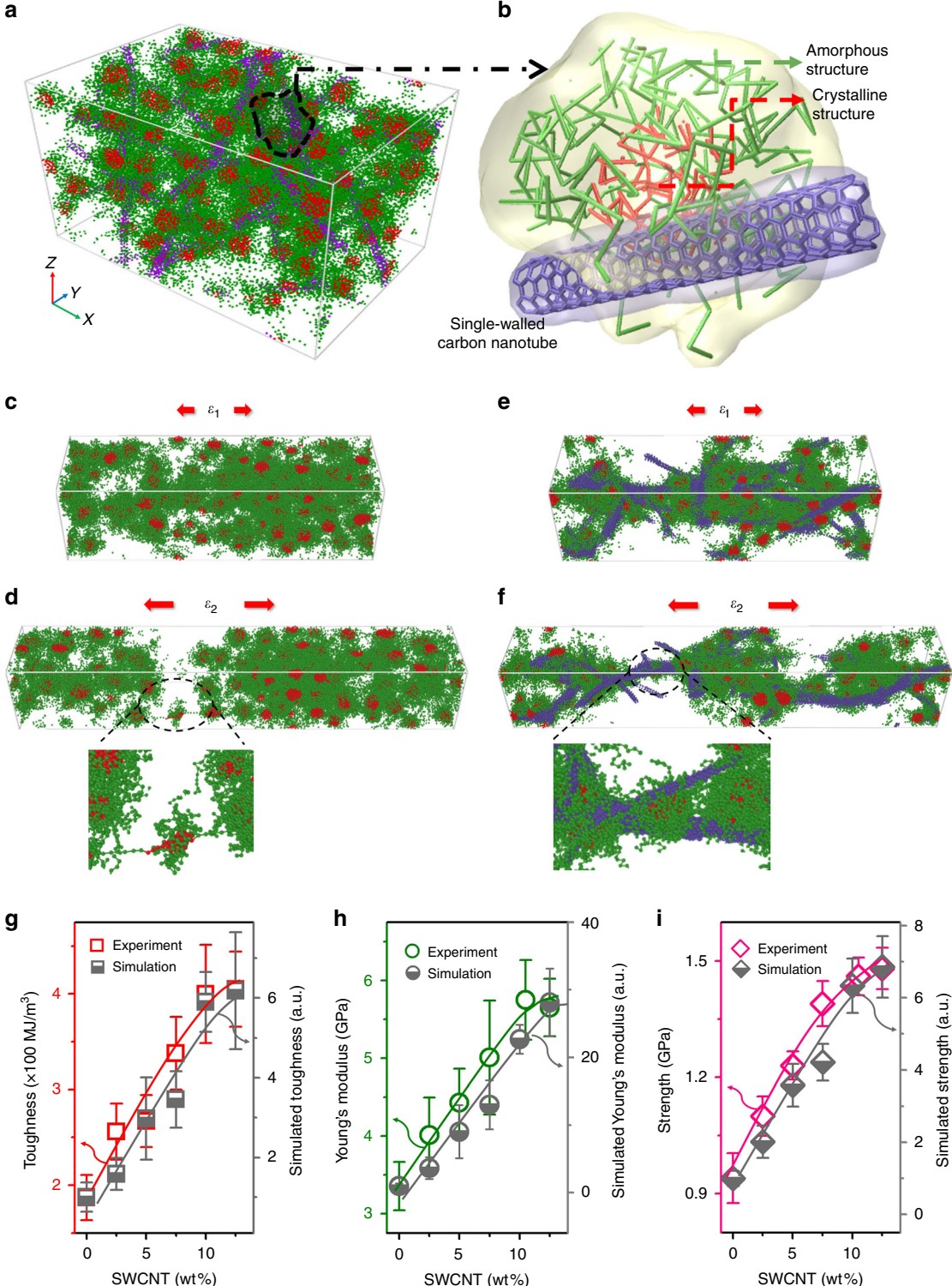

**Fig. 2 A typical mechanical test of the silk-SWCNT nanocomposite in DPD simulation. a, b** Show a snapshot of the coarse-grained S-silk composite. Red: crystalline structure, beta-sheet structure; green: amorphous structure ($3_1$-helices and beta-turns); purple: SWCNT. **c–f** DPD-simulated images showing the structural evolution of natural S-silk (**c, d**) S-silk composite@10% SWCNT (**e, f**). Along the x-axis with increasing strain, $\varepsilon_1 < \varepsilon_2$. $\varepsilon_2$ is the critical strain at which spider silk is broken. Because of the hydrophobic interactions between SWCNT and spider silk, the S-silk composite has better mechanical properties than natural S-silk. **g–i** Graph shows Toughness, Young's modulus and strength increased with increasing wt% of SWCNT, indicating that SWCNT is critical for improving the mechanical properties of the S-silk composite. DPD simulation and experiments agree well. The error bars of **g–i** for experiments show standard deviations based on 50 independent samples. The error bars of **g–i** for simulations show standard deviations based on 5 independent simulations.

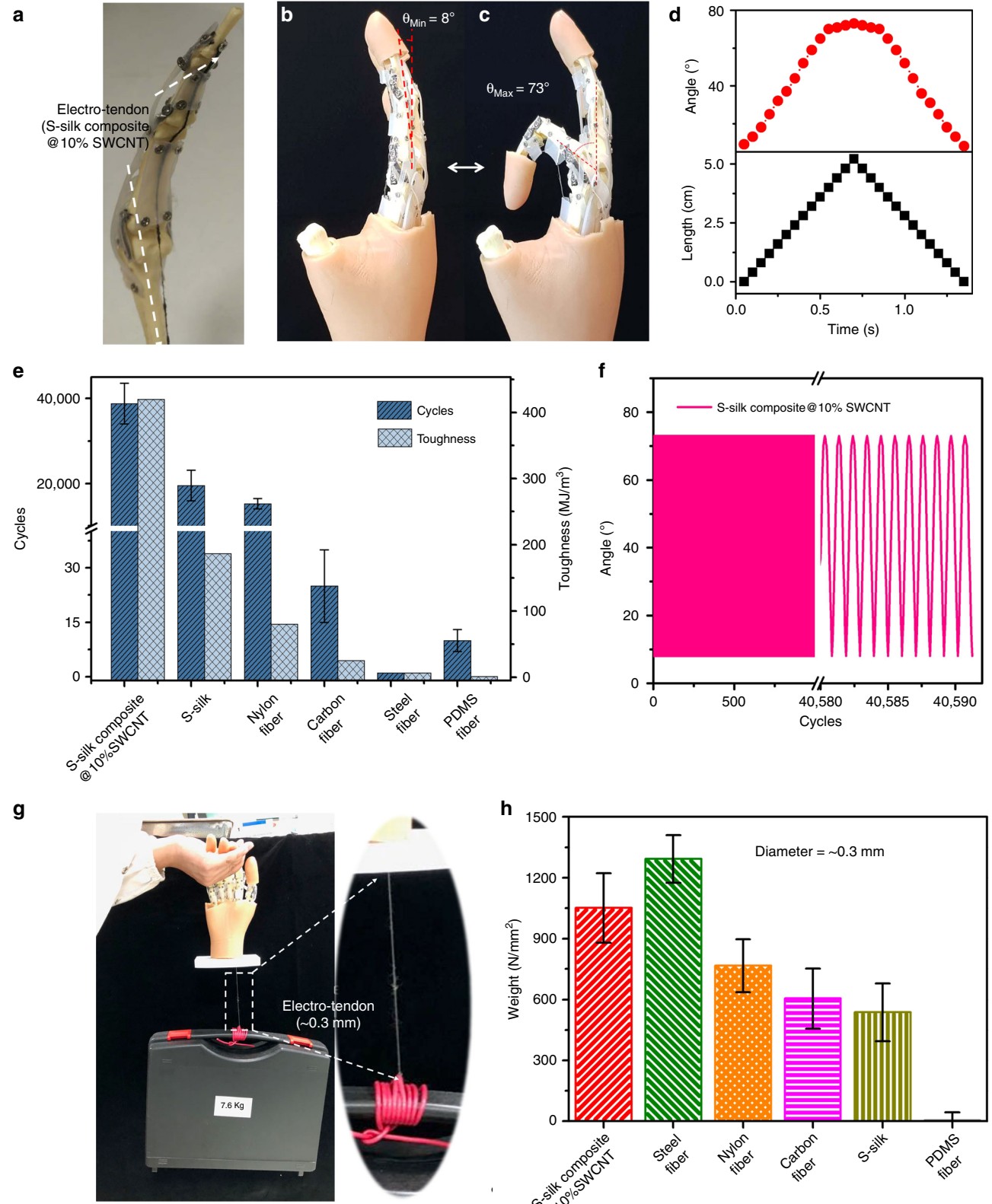

based on S-silk composite with toughness of 420 MJ/m³ sustained 40,000 cycles of the full bending process (Fig. 3f), nearly double that of natural S-silk with toughness of ~190 MJ/m³. For nylon and carbon fibers with lower toughness, the endurance of these robotic hands was considerably lower. In the case of steel fiber and PDMS fiber, the finger failed to perform a full bending action due to poor toughness (Supplementary Fig. 12).

We further tested the ability of the fingers to lift a weight; diameter of all fibers was kept at 0.3 mm. The finger with the S-silk composite fiber can lift a weight of 7.6 kg (Fig. 3g) (1051.1 N/mm²), which was comparable to the finger with steel fiber (1292.5 N/mm²) but much higher than fingers with nylon fiber (766.3 N/mm²), commercial carbon fiber (604.7 N/mm²), natural S-silk (537.1 kg/mm²) and PDMS fiber (4.3 N/mm²) (Fig. 3h).

**Fig. 3 Performance of humanoid robotic hands assembled with S-silk composite as electro-tendon. a** Photograph of a 3D-printed robotic finger with an S-silk composite@10% SWCNT electro-tendon held in place by a silicone-based extensor. **b** Photograph showing the index finger at resting state has an angle of about 8°. **c** Photograph of the finger bent to the maximum position, pulled by the tendon. **d** Graphs show the bending angle (red curve) of the robotic finger correlates with the change in length (black curve) of the electro-tendon. The angle at maximum bending is 73° (peak of the red curve). This whole process has taken ~1.5 s. **e** Cyclic bending tests of robotic fingers assembled using different materials as the tendon show tougher materials were more durable. Our S-silk@10% SWCNT could withstand ~40,000 cycles of the full bending process, nearly double that of fingers using natural S-silk and nylon fiber. Because of low toughness, fingers using steel fiber and PDMS fiber failed to complete a full bending process. The error bars of **e** show standard deviations based on 10 independent samples. **f** Graph shows the endurance of the finger using S-silk composite @10% SWCNT. **g**, **h** Lifting weight of humanoid robotic finger using different types of fibers (diameter of all fiber is 0.3 mm). S-silk composite@10%SWCNT loaded about 7.6 kg, which was comparable to steel fiber. The error bars of **h** show standard deviations based on 10 independent samples.

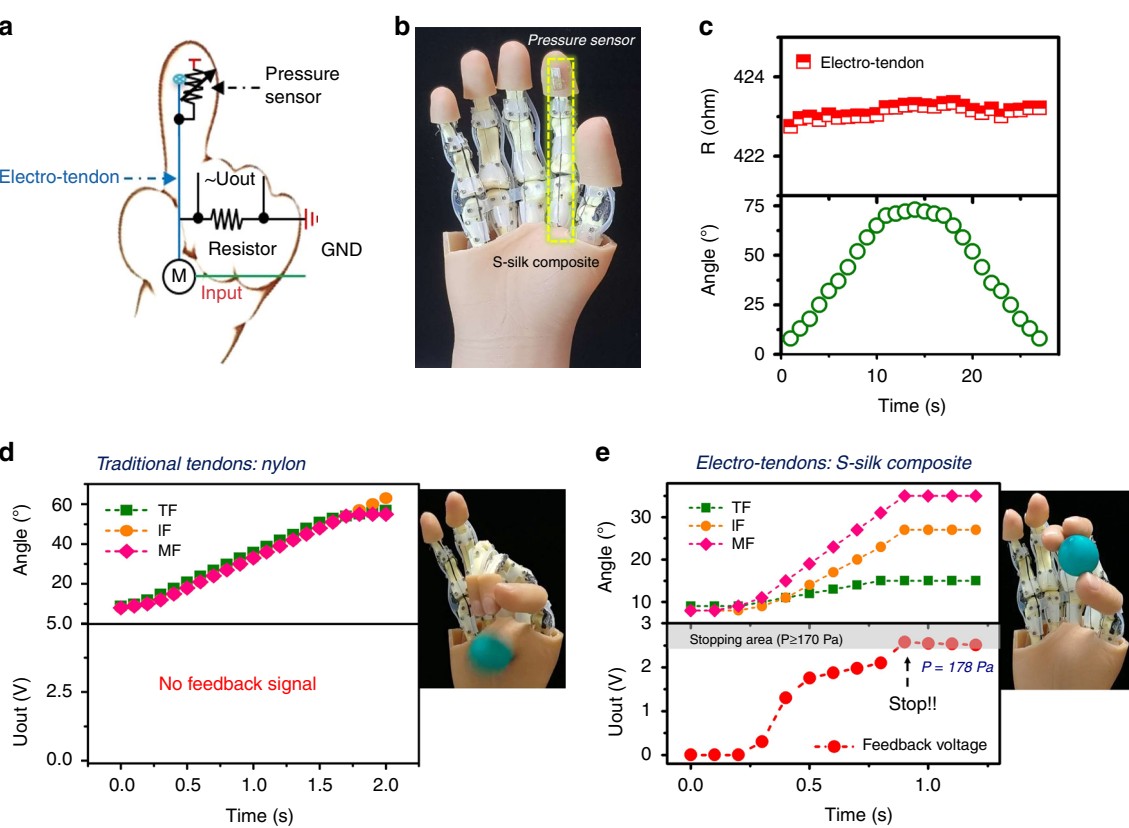

**Fig. 4 Feedback processes of the humanoid robotic hand when grasping objects. a** Schematic of the actuating layout and sensing circuit for the humanoid robotic hand. S-silk composite@10% SWCNT was used as the electro-tendon to transmit force from the servo motor of a transmission system and electrical signal from the pressure sensor of the sensing system. Here $U_{out}$ is the voltage of a reference resistor. Input is the program command of a software that depends on $U_{out}$. M: stepping motor; GND: ground. **b** Photo of the 3D-printed humanoid robotic hand assembled with an S-silk composite@10% SWCNT electro-tendon (yellow dotted rectangle) and a pressure sensor mounted on the index finger. **c** Graphs show the resistance (red curve) of the electro-tendon did not change when the finger was bent at different angles (green curve). **d** Grasping process of the robotic hand based on traditional tendon based on nylon fiber. Because nylon is an insulator, no signal is transmitted from the pressure sensor to the software, preventing the finger from catching the green balloon. TF: thumb finger; IF: index finger; MF: middle finger. **e** Grasping process of the robotic hand based on the S-silk composite@10% SWCNT electro-tendon. The software is programmed to stop the finger when $P \geq 170$ Pa (gray area). Because the electro-tendon has high and stable conductivity, the electrical signal from the pressure sensor can be transmitted to the software, which directs the servo motor to respond by pulling the electro-tendon and bending the finger to the appropriate angle based on the pressure data to capture the green balloon.

**Feedback and grasping process of robotic hand based on electro-tendon.** Besides endurance, electro-tendons also need to stably transfer signals from the feedback system during movement of the robotic hand. To achieve this, we designed a pressure feedback system (Fig. 4a and Supplementary Fig. 13) to enable the hand to feel the pressure of the finger when grasping objects. The pressure sensor, which is based on the pyramidal structure we reported previously[45], has a sensitivity of about 24.8 kPa⁻¹. This sensor can detect pressures from 0 to 1 kPa in <4 ms, which is enough for our grasping experiments. We assembled the pressure sensor on the tip of the index finger and connected it to the

electro-tendon with a reference resistor of 100 kΩ as shown in Fig. 4b. When the finger was bent, the electrical resistance of the electro-tendon remained nearly unchanged (Fig. 4c). When we touched the sensor with pressures of 0, 113, 327, and 749 kPa, the finger bent by 0°, 19°, 32° and 43°, respectively. Higher forces resulted in greater bending angles (Supplementary Fig. 14 and Supplementary Movie 4).

We used the humanoid robotic hand assembled with the S-silk composite electro-tendon and pressure sensor to grasp a green balloon without deforming its shape. A tendon based on nylon, a common non-conductive material for robotic tendon, was used

for comparison. After placing the balloon in the hand, the actuating system was activated to bend the fingers until the index finger touched the balloon and recorded a pressure value that fulfills the criterion of the programmed stop pressure (in this case, the stop pressure was ≥170 Pa, determined through trial and error). When the stop pressure criterion is met, the fingers stop bending, allowing the green balloon to remain in place between the three fingers (Fig. 4d and Supplementary Movie 5). For this balloon, which has a diameter of 4.8 cm, the pressure sensor on the index finger recorded a pressure of 178 Pa. In the case of the robotic hand with the non-conductive nylon, the hand continued to bend forward and failed to grasp the balloon because no signal was transmitted from the pressure feedback system to halt the hand. (Fig. 4e and Supplementary Movie 5). The humanoid hand assembled with S-silk composite electro-tendons could also grasp other objects such as a needle and a puff (Supplementary Fig. 15). In the case of the needle, the hand stopped moving when the force detected by the pressure sensor was over 0.012 N (in this case, the force recorded was 0.015 N). Because the touch area between the needle and the sensor cannot be defined, here, we used the force detected by the sensor when the hand grasped the needle as reference. In the case of the puff, the hand was programmed to stop moving when the pressure was above 400 Pa (in the experiment, the pressure recorded was 430 Pa). This robotic hand based on the electro-tendon is clearly dexterous enough to perform basic grasping functions that are useful for day-to-day activities.

## Discussion

In conclusion, we report the fabrication of a high toughness electro-tendon that enabled a humanoid robotic hand to perform basic grasping functions. The electro-tendon is a flexible and conductive composite material made from spider silk, coated with a conductive PEDOT:PSS layer and toughened with SWCNT. It is tough and durable, able to withstand up to 40,000 cycles of bending and stretching without any change in conductivity. When attached to a pressure sensor and mounted on a humanoid robotic finger, the robotic hand could grasp various objects such as a balloon, needle and a puff without crushing and damaging the objects. The electro-tendon, which can transmit force from the servo motor of the actuation system and electrical signals from the pressure sensor of the feedback system, simplifies the setup of a humanoid robotic finger because both actuation and sensing signals can now be transmitted through a single fiber. This material, which imparts improved dexterity and durability to the robotic hand, is expected to have many applications, particularly in the building of robots that could help with household chores such as picking up and putting away clutter. Besides robotics, this super tough conductor could also serve as an electrode for interconnector for highly tough flexible electronic circuits, anti-static durable woven and tough cables.

## Methods

**Feeding the spider and collection of the spider silk**. The spider was kept in an $80 \times 60 \times 40$ cm vivarium, consisting of wood panels and artificial plants (shown in Supplementary Fig. 1 and Supplementary Movie 1). The spider was kept at a humidity above 65% and a temperature about 25 °C. The spider was fed live locusts and flies three times a week. We collected the spider silk every 2 weeks using a scalpel for transfer within a rigid frame. Herein, we used a classic spider silk, dragline, as an example. The diameter was 3–4 μm (Supplementary Fig. 1b and c), and the surface was smooth.

**Fabrication of the spider silk composite**. All processing solvents and chemicals, such as ethanol, silver nitrate (AgNO₃), 7,7,8,8-Tetracyanoquinodimethane (TCNQ), and poly(3,4-ethylenedioxythiophene) poly(styrenesulfonate) (PEDOT: PSS) were analytic grade reagents and purchased from Sigma-Aldrich. The SWCNTs were purchased from Nanjing XFNANO Materials Tech Co, Ltd.

**1. Preparation of nano-island structure on the spider silk**. Firstly, the collected raw spider silk was rinsed by ethanol (Absolute, 99.9%) three times and dried at 80 °C about 12 h. Then, the spider silk was hydrophilization by the plasma under O₂ atmosphere for 10 min, forming hydrophilic groups, such as hydroxyl groups on the surface of spider silk (Supplementary Fig. 2a). Secondly, the modified spider silk was immersed in 0.1 mol/L AgNO₃ of ethanol for 5 min. The hydrophilic group −OH was changed into −OAg as the seeds for nano-island structure growth (Supplementary Fig. 2b). Thirdly, the spider silk-OAg was immersed in 0.01 mol/L tetracyanoquinodimethane (TCNQ) of ethanol through the method of drop flowing and annealed (5 min, 100 °C) to form a layer of the nano-island structure. Detailly, the spider silk-OAg was laid along with the beaker. The 0.01 mol/L TCNQ dropped from the funnel and flowed past the spider silk-OAg. And the flow rate was about 1 drop per 1 s. The whole process was about 10 min. After that, the modified spider silk-OAg was annealed for 5 min at 100 °C. At last, a nano-island structure formed on the surface of spider silk (Supplementary Fig. 2c).

**2. Preparation of conductive spider silk composite**. We prepared PEDOT:PSS solution with 2.5%, 5%, 7.5%, 10%, and 12.5% SWNCT. And then, the spider silk of (1) was immersed in the PEDOT:PSS solution also via the method of drop flowing. The rate was 1 drop per 5 s. After annealing at 80 °C for 5 min, the diameter changed from 3–4 to 6–7 μm and a conducting wrinkle structure was formed on the surface of spider silk which was very helpful for unchanged conductivity during stretching/compression. The reason for the wrinkle structure was the intrinsic shrinkage of spider silk induced by the solution of water in PEDOT:PSS (Supplementary Fig. 2d).

**3. Characterization of morphology, mechanical and electrical properties of spider silk composite**. We used field emission scanning electron microscope (SEM) to characterize the morphology, cross-section of the spider silk composite and microstructural evolution of the spider silk under different strain. SEM images shown in Supplementary Fig. 4 were obtained on a Zeiss SUPRA55 SEM using an acceleration voltage of 5 kV in situ at the same area from strain at 0% to breakage using the same magnifications. Cross-section images (Fig. 1d and Supplementary Fig. 1c) were obtained by cutting the encapsulation epoxy resins with spider silk along the silk direction using a sharp blade, after freezing the fiber in liquid nitrogen. More detailly, the spider silk composite was embedded in a drop of encapsulation epoxy resin which was solidified under UV-light for 30 min. Then, we used a sharp blade to cut the resin along the silk direction for SEM under liquid nitrogen. The mechanical properties of spider silk and its composite were obtained using an Instron mechanical tester (MTS criterion Model 42) with 50 N load cell and 100 N Bionix vice grips. Stress–strain curves for non-coating and coating spider silk were shown in Supplementary Fig. 4. The Young's modulus of silk was calculated by linear curve fitting for strain below 15%. The toughness was an area of the stress–strain before breakage. The electrical properties of the samples obtained by two-probe resistance measurements using Keithley 4200-SCS semiconductor characterization system. Stretch/compress was applied along the axial direction of the spider silk using a homemade fixture. The resistance of the spider silk composite nearly unchanged until the strain at 60%. This reason was the conductive path unchanged when the wrinkle structure of the conductive layer PEDOT:PSS@SWCNT gradually flattened with increasing of the strain shown in the blue area in Supplementary Fig. 6. And, we could see clearly there was no creak even at the strain at 50%. But the sample was breakage during the strain increased to 60%.

**Dissipative particle dynamic (DPD) simulation models and method**. DPD is a mesoscopic particle-based coarse-graining (CG) simulation method[46,47]. Comparing with the all-atom simulation method, DPD is advantageous to treat a much larger spatial scale and time scale in simulations. For example, the DPD method is well used to simulate the interaction of carbon materials with cellular membrane[48,49], the deformation of carbon nanotubes[50], the morphology and dynamics of carbon nanotube in polycarbonate carbon nanotube composite[51] and etc. In this study, we adopted the similar DPD model developed by Lin et al. for the spider silk protein, which is well applied to study the assembly and deformation of silk protein[43]. In the DPD simulations, interaction forces between beads include $\mathbf{F}_{ij}{}^C$, $\mathbf{F}_{ij}{}^D$, and $\mathbf{F}_{ij}{}^R$, denoting the conservative force, the dissipative force, and the random force, respectively. The total force on bead $i$ is given by

$$\mathbf{F}_i = \sum_{i \neq j} \left( \mathbf{F}_{ij}^C + \mathbf{F}_{ij}^D + \mathbf{F}_{ij}^R \right) \tag{1}$$

where

$$\begin{aligned} \mathbf{F}_{ij}^C &= a_{ij}\omega(r_{ij})\hat{\mathbf{r}}_{ij} \\ \mathbf{F}_{ij}^D &= -\gamma\omega^2(r_{ij})(\mathbf{v}_{ij} \cdot \hat{\mathbf{r}}_{ij})\hat{\mathbf{r}}_{ij} \\ \mathbf{F}_{ij}^R &= \sigma\omega(r_{ij})\xi_{ij}\Delta t^{-1/2}\hat{\mathbf{r}}_{ij} \end{aligned} \tag{2}$$

In Eq. (2), $r_{ij}$ is the distance, $\hat{\mathbf{r}}_{ij}$ is the unit vector, and $\mathbf{v}_{ij}$ is the relative velocity between beads $i$ and $j$. $\gamma$ and $\sigma$ are the parameters with the relation as $\sigma^2 = 2\gamma k_B T$, where $k_B$ is the Boltzmann's constant and $T$ is the temperature. $\sigma = 3.0$ and $\gamma = 4.5$ are used as the previous study[43,44]. $\xi_{ij}$ is a normal distribution number with zero

**Table 1 DPD parameters for the CG beads.**

|        | "a" | "b" | "w" | "c" |
|--------|-----|-----|-----|-----|
| "a"    | 25  | 27  | 40  | 25  |
| "b"    | 27  | 25  | 26  | 27  |
| "w"    | 40  | 26  | 25  | 40  |
| "c"    | 25  | 27  | 40  | 25  |

mean and unit variance. $\omega(r_{ij})$ is the normalized distribution function. The hydrophilic and hydrophobic properties of the CG beads are defined by the parameter $a_{ij}$ in Eq. (2)[52].

The DPD potential parameters for the CG beads of water molecules and silk peptides were taken from Lin et al.'s work[43]. To represent the hydrophobicity of SWCNT, the DPD potential parameter for bead "c" in SWCNT is taken as the same as bead "a", according to previous study[53]. All $a_{ij}$ values are listed in Table 1[43,53].

The DPD simulations were carried out by LAMMPS package[54]. For the starting structure, the silk peptide chains were randomly generated in a rectangular simulation box with a dimension of $60 \times 40 \times 40\ r_c^3$. The simulation box is under periodic boundary condition. To simulate varied weight fraction of SWCNT, the proper number of SWCNT were put into the simulation box in random position and direction. After building up the initial structure, equilibration simulation was performed until a steady state is achieved. Then the entire simulation box containing silk peptides, SWCNT and solvent are stretched in the x-axis at a constant engineering strain rate of $5 \times 10^{-5}\tau^{-1}$, while the y and z dimensions are adjusted simultaneously (with their aspect ratio fixed) to maintain the constant volume of the simulation box[43]. All DPD simulations run under the NVE ensemble.

The Young's modulus of the nanocomposite in the x-axis, $E_x$, can be obtained from the three-dimensional system by the equation: $E_x = [\sigma_x - \upsilon(\sigma_y + \sigma_z)]/\varepsilon_x$, where $\sigma_x$, $\sigma_y$, and $\sigma_z$ are the normal stress tensor value along the corresponding axis, and the Poisson's ratio $\upsilon = 0.5$ for the constant volume constraint[43]. The effective stress can be then defined as $\sigma_{eff} = \sigma_x - \upsilon(\sigma_y + \sigma_z)$ and the region (e.g. $\varepsilon \leq 25\%$) of the stress–strain curve as linear-elastic and used to determine $E_x$[43]. Supplementary Fig. 8b shows a typical mechanical test of the silk–SWCNT nanocomposite in DPD simulation. Each structure was repeated five times to determine its mechanical properties.

It should be noted that as discussed by Lin et al.[43], due to the coarse-grained nature of the DPD model, the absolute stress values are approximate; and in addition to the presence of solvents in simulations, would also lead to simulated ultimate strains much higher than that of common silk fibers. In the DPD simulations, we will focus on the relative values in a qualitative way and the mechanism of how SWCNT enhance the mechanical properties of the silk fibers.

## Data availability
The authors declare that the main data supporting the findings of this study are available within the article and its Supplementary Information files. Extra data are available from the corresponding author upon request.

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

## Acknowledgements

This work was supported by the National Research Foundation (NRF), Prime Minister's office, Singapore, under its NRF Investigatorship (NRF2016NRF-NRF1001-21) and Singapore Ministry of Education (MOE2017-T2-2-107), National Natural Science Foundation of China (NSFC 11932017, 11772054, 11772055, and 11532009). We also gratefully acknowledge the financial support from the Agency for Science, Technology and Research (A*STAR) through Advanced Manufacturing and Engineering (AME) Programmatic Grant (No. A19A1b0045) and the use of computing resources at the A*STAR Computational Resource Centre, and National Supercomputing Centre, Singapore. We also thank Ai Lin Chun for critically reading the manuscript.

## Author contributions

L.P. and X.C. conceived the idea. L.P. prepared and characterized the sample. L.P., W.R. L., and M.W. conducted the mechanical and electrical measurements. F.W., Y.C., Y.-W. Z., B.J., and D.L. discussed and performed the dissipative particle dynamic (DPD) simulation. L.P. and P.C. designed robotic hands and conducted grasping process. L.P., F.W., D.L., and X.C. co-wrote the paper. All the author discussed the results and commented on the manuscript.

## Competing interests

The authors declare no competing interests.
