## [Peer Review File · Nature Communications]

Reviewers' comments:

Reviewer #1 (Remarks to the Author):

Pan and coauthors fabricated an electro-tendon, which is a composite material made from spider silk. The spider silk that the authors used has high toughness, and this was further improved by coating PEDOT:PSS@SWCNT through the drop flowing method. The fabricated s-silk composite had high toughness, conductivity, and stretchability, suitable material properties enough to be used for stretchable electronics. Applied to a humanoid robotic hand, the electro-tendon was able to simultaneously transmit force from the actuation system and electrical signals from the pressure sensor. This enabled the feedback system, allowing the robotic hand to stably grasp objects by applying the suitable force. This work gives a new alternative for developing the flexible and stretchable conductor. Most of current researches on this area focuses on developing nanocomposite materials by combining synthetic materials, such as conductive polymers, liquid metals, nanoparticles, and nanowires. However, in this work, it is noteworthy that existing natural material was used, and it was further improved through nanomaterials. It also gives inspiration to many applications, particularly to the field of soft electronics and wearable devices. Therefore, the reviewer recommends publication of this manuscript in Nature Communications. Some minor issues that need to be addressed before publication are as follows.

Comment #1: The authors showed that S-silk composite's toughness was maintained under 40,000 cycles of the repetitive full bending process. The reviewer agrees that the S-silk composite has surprising toughness. However, the reviewer also wonders whether the conductivity changes under such intensive circumstances (not only 1,000 cycles). Please provide conductivity change data.

Comment #2: The authors showed that the S-silk could withstand a load of 7.6kg. However, to practically operate as the artificial "tendon", it should actually be able to lift the load, against the gravity. The reviewer suggests authors show how much weight this tendon can lift.

Comment #3: The reviewer guesses that the S-silk tendon will be stretched over 15% when it withstands (or overcome) some heavy weights (unlike human tendon). In that situation, conductivity will also be changed. Isn't there any impact of this conductivity changes on manipulating the robotic hand when transmitting signals?

Comment #4: Authors realized the humanoid robotic hand that can grasp objects. In movie 5, three fingers are involved in this process to grasp the balloon. In case of the index finger, it is possible to apply proper pressure using the feedback system. For other fingers, however, it is not clearly understandable how this could be possible, as it seems that there isn't any pressure sensor on these fingers. Please explain how the three fingers worked in harmony to grasp the object.

Reviewer #2 (Remarks to the Author):

Comments

Chen et al reported a very interesting electro-tendon based on spider silk with a toughness of 420 MJ/m³ and conductivity of 1077 S/cm that integrated the roles of wire and tendon to transmit electrical signals and force. This electro-tendon modified by SWCNT and PEDOT:PSS and can bend/stretch more than 40000 cycles with stable conductivity. Because of these properties, they have mounted it on a robotic finger with pressure sensor and made the robotic hand grasp different things without additional circuit components. I think that this work has profound significance for simplifying

the complex circuit of robots and expanding the practical application of spider silk. Owing to these merits of this manuscript, I can strongly recommend the publication of this manuscript in Nature Communications, after the authors make some following revisions.

[Major]

1 In Figure 1, the authors have shown the optical image of native spider silk. For comparison, please show us the optical image of modified spider silk.

2 In the manuscript, the authors have shown the mechanical properties of the spider silk composite such as toughness, Young's modulus and strength was enhanced after introducing SWCNT. As an important parameter of stretchable electrode, did the maximum strain of spider silk composite changed? And how? I think the authors should show the relevant results.

3 The different type of spider silk shows various mechanical properties. In this manuscript, the authors have used dragline silk for research. How about other type of spider silk? Do they have similar phenomenon after introducing SWCNT? And can they use as the electro-tendon?

4 What's the role of PEDOT:PSS for fabricating spider silk composite? Because the SWCNT also is a good conductive material, can only SWCNT be used for improving conductive of spider silk?

5 Could the authors provide more detailed explanation on why to choose the DPD model in this system, and comment on robust is this model?

6 In the DPD simulations, the authors represented a single silk peptide by a sequence as 16b-5a-7b-5a-7b. Did the sequence affect the simulation results?

7 In Figure 3d, a full bending process of robotic finger costed about 25 s. It is too long. For a human, this process may be 1-2s. Can the robot finger do the bending process in this time?

8 As the manuscript mentioned, the strength of the spider silk composite@10% SWCNT was about 1.5 GP which was better than normal steel fiber. But why the Loading weight of spider silk composite@10% SWCNT was lower than steel fiber in Figure 3h?

[other Minor issues]

1 In Figure 1g, the author should correct the unit (%) to (wt%).

2 In Figure 3h, there is a misunderstand. The author should correct the unit (kg) to (kg/m²).

Detailed responses to reviewers' comments

Reviewer #1

Comment: *Pan and coauthors fabricated an electro-tendon, which is a composite material made from spider silk. The spider silk that the authors used has high toughness, and this was further improved by coating PEDOT:PSS@SWCNT through the drop flowing method. The fabricated s-silk composite had high toughness, conductivity, and stretchability, suitable material properties enough to be used for stretchable electronics. Applied to a humanoid robotic hand, the electro-tendon was able to simultaneously transmit force from the actuation system and electrical signals from the pressure sensor. This enabled the feedback system, allowing the robotic hand to stably grasp objects by applying the suitable force. This work gives a new alternative for developing the flexible and stretchable conductor. Most of current researches on this area focuses on developing nanocomposite materials by combining synthetic materials, such as conductive polymers, liquid metals, nanoparticles, and nanowires. However, in this work, it is noteworthy that existing natural material was used, and it was further improved through nanomaterials. It also gives inspiration to many applications, particularly to the field of soft electronics and wearable devices. Therefore, the reviewer recommends publication of this manuscript in Nature Communications. Some minor issues that need to be addressed before publication are as follows.*

Response: We thank the reviewer for the positive comment on our work. We have carefully considered the comments and revised the manuscript accordingly as described below.

Q#1: *The authors showed that S-silk composite's toughness was maintained under 40,000 cycles of the repetitive full bending process. The reviewer agrees that the S-silk composite has surprising toughness. However, the reviewer also wonders whether the conductivity changes under such intensive circumstances (not only 1,000 cycles). Please provide conductivity change data.*

Response: We have measured the conductivity of the S-silk composite under more than 36,500 cycles. The conductivity (~1.1 k S/cm) was nearly unchanged under strain from 0 to 15%. Related revised figure is shown in the Supplementary Figure 7a and related discussion is added in the revised manuscript on page 5 line 24-27.

Q#2: *The authors showed that the S-silk could withstand a load of 7.6 kg. However, to practically operate as the artificial "tendon", it should actually be able to lift the load, against the gravity. The reviewer suggests authors show how much weight this tendon can lift.*

Response: We have corrected the Figure 3h in our revised manuscript using the unit of N/mm². Related discussion is added in the revised manuscript on page 7 line 24-29.

Q #3: *The reviewer guesses that the S-silk tendon will be stretched over 15% when it withstands (or overcome) some heavy weights (unlike human tendon). In that situation, conductivity will also be changed. Isn't there any impact of this conductivity changes on manipulating the robotic hand when transmitting signals?*

Response: We agree with the reviewer that the S-silk tendon will over 15% when it withstands some heavy weights. Also, from the Supplementary Figure 5b, the conductivity changed

about 5% when the strain increased to 60%. Before the rupture of the S-silk composite, the change rate ~5% in conductivity has little impact on manipulating the robotic hand when transmitting signals. But, the strain ~60% of tendon will induce broken of function of the robotic hands such as grasping. Thus, in our work, we measured the changes of conductivity with strain from 0% to 15%. Related discussion is also added in the revised manuscript on page 5 line 27-29.

Q#4: *Authors realized the humanoid robotic hand that can grasp objects. In movie 5, three fingers are involved in this process to grasp the balloon. In case of the index finger, it is possible to apply proper pressure using the feedback system. For other fingers, however, it is not clearly understandable how this could be possible, as it seems that there isn't any pressure sensor on these fingers. Please explain how the three fingers worked in harmony to grasp the object.*

Response: Actually, every activity of our robotic hand such as grasping the balloon needs a lot of training processes. Firstly, we manual stop the hands without pressure sensor when it grasped the balloon. During these training processes, we figured out the appropriate bending speed of every finger such as thumb, index finger and middle finger when the robotic hand harmoniously grasped the balloon. After we achieved the bending speed, we assembled the pressure sensor on the figure to measure the pressure. Then, we used the value of the pressure as a reference to programmed control, instead of manual control, the stopping of the hands when it harmoniously grasped the balloon. In the case of manuscript, the pressure of index finger was $\geq 170\text{Pa}$. Thus, we don't need assemble the pressure sensor on every finger. Certainly, we can assemble the pressure on three fingers to control the robotic hand to grasp the object. However, multi-controlling channels are more complex than one controlling channel when considered synchronous controlling. In addition, in our manuscript, we are more focus on the intrinsic properties of the electro-tendon. In future engineering application, we will assemble more various sensors on the different fingers for more functions. Related discussion is also added in the revised supporting information on page S19-S20.

Reviewer 2#

Comment: *Chen et al reported a very interesting electro-tendon based on spider silk with a toughness of 420 MJ/m³ and conductivity of 1077 S/cm that integrated the roles of wire and tendon to transmit electrical signals and force. This electro-tendon modified by SWCNT and PEDOT:PSS and can bend/stretch more than 40000 cycles with stable conductivity. Because of these properties, they have mounted it on a robotic finger with pressure sensor and made the robotic hand grasp different things without additional circuit components. I think that this work has profound significance for simplifying the complex circuit of robots and expanding the practical application of spider silk. Owing to these merits of this manuscript, I can strongly recommend the publication of this manuscript in Nature Communications, after the authors make some following revisions.*

Response: We thank the reviewer for the positive comment on our work. We have carefully considered the comments and revised the manuscript accordingly as described below.

Q#1: *In Figure 1, the authors have shown the optical image of native spider silk. For comparison, please show us the optical image of modified spider silk.*

Response: As comparison, the optical image of modified spider silk (S-silk composite@ 10% SWCNT) shown in Supplementary Figure 3. The color of the modified spider silk changed into black. Related discussion is also added in the revised manuscript on page 5 line 2-4.

Q#2: *In the manuscript, the authors have shown the mechanical properties of the spider silk composite such as toughness, Young's modulus and strength was enhanced after introducing SWCNT. As an important parameter of stretchable electrode, did the maximum strain of spider silk composite changed? And how? I think the authors should show the relevant results.*

Response: We measured the changes of maximum strain with increasing SWCNT content. From Supplementary Figure 6, the maximum was nearly unchanged along the SWCNT content. Related discussion is also added in the revised manuscript on page 5 line 19-20.

Q#3: *The different type of spider silk shows various mechanical properties. In this manuscript, the authors have used dragline silk for research. How about other type of spider silk? Do they have similar phenomenon after introducing SWCNT? And can they use as the electro-tendon?*

Response: We agree with reviewer that the different type of spider silk shows different mechanical properties. There are three types of major ampullate silk such as dragline silk, radius silk and frame silk. The dragline silk shows better mechanical properties than other two silks. Thus, in our work, we used the dragline silk to fabricated electro-tendon. Actually, the phenomenon in this work is also according with other types of spider silk. Here, we found that the toughness and conductivity of radius silk also improved with increasing of SWCNT content. The radius silk@ 10% SWCNT shows conductivity of 1051 S/cm and toughness of 227 MJ/m³ which also can be used as the electro-tendon, as shown in the following figure.

Figure R1 (a,b) Conductivity and toughness of the radius silk composite increased with increasing weight percent of SWCNT, before experiencing saturation at 12.5wt%. The maximum conductivity and toughness achieved were 1051 S/cm and 227 MJ/m³ which also can be used as the electro-tendon

Q#4: *What's the role of PEDOT:PSS for fabricating spider silk composite? Because the SWCNT also is a good conductive material, can only SWCNT be used for improving conductive of spider silk?*

Response: PEDOT:PSS have two roles in our work. The first role is the dispersant. Because, the SWCNT is nanoscale powder and it can disperse well in PEDOT:PSS. The second role is that PEDOT:PSS is flexible organic with excellent electrical conductivity which can greatly improve the conductivity of spider silk. Related discussion is also described in the revised manuscript on page 4 line 22-26.

Q#5: *Could the authors provide more detailed explanation on why to choose the DPD model in this system, and comment on robust is this model?*

Response: The dissipative particle dynamic simulation method is a mesoscopic particle-based coarse-graining (CG) simulation approach. Comparing with the all-atom simulation method, DPD is advanced to treat a much larger spatial scale and time scale in simulations. For example, the DPD method is well used to simulate the interaction of carbon materials with cellular membrane [1, 2], the deformation of carbon nanotubes [3], the morphology and dynamics of carbon nanotube in polycarbonate carbon nanotube composite [4] and etc. In the recent papers by Markus J. Buehler and co-workers, a DPD model for the spider silk protein was developed, which is well applied to study the assembly and deformation of silk protein [5, 6]. So that in the silk-SWCNT system of this study, the DPD method enables us to study the interaction between the silk-protein and SWCNTs, and simulate the mechanical properties of the structures. Related discussion is also added in the revised manuscript on page 12 line 4-11 and related rf. from [46]-[51].

References:

- [1] Höfner S, Melle-Franco M, Gallo T, et al. A computational analysis of the insertion of carbon nanotubes into cellular membranes[J]. *Biomaterials*, 2011, 32(29): 7079-7085.
- [2] Q. C. Wang, X. B. Zhai, M. Crowe, L. Gou, Y. F. Li, D. C. Li, L. Zhang, J. J. Diao, and B. H. Ji, Heterogeneous oxidization of graphene nanosheets damages membrane[J], *Sci.*

China-Phys. Mech. Astron., 2019, 62: 064611.

- [3] Liba O, Kauzlarić D, Abrams Z R, et al. A dissipative particle dynamics model of carbon nanotubes[J]. Molecular Simulation, 2008, 34(8): 737-748.
- [4] Chakraborty S, Choudhury C K, Roy S. Morphology and dynamics of carbon nanotube in polycarbonate carbon nanotube composite from dissipative particle dynamics simulation[J]. Macromolecules, 2013, 46(9): 3631-3638.
- [5] Lin, S., Ryu, S., Tokareva, O., Gronau, G., Jacobsen, M. M., Huang, W., et al. (2015). Predictive modelling-based design and experiments for synthesis and spinning of bioinspired silk fibres. Nature Communications, 6, 6892.
- [6] Rim N G, Roberts E G, Ebrahimi D, et al. Predicting silk fiber mechanical properties through multiscale simulation and protein design[J]. ACS biomaterials science & engineering, 2017, 3(8): 1542-1556.

Q#6: *In the DPD simulations, the authors represented a single silk peptide by a sequence as 16b-5a-7b-5a-7b. Did the sequence affect the simulation results?*

Response: We used the same sequence as the previous study (Lin et al., Nature Comm. 2015) to mimic the spider silk, which showed that the above sequence well reproduced the interaction network of the silk protein. The sequence is to mimic the hydrophobic/hydrophilic nature of the silk protein as a copolymer. As long as the sequence represented the silk protein as hydrophobic blocks (e.g., the beads “a”) interlude by hydrophilic ones (e.g., the beads “b”), it will not affect the qualitative results.

Q#7: *In Figure 3d, a full bending process of robotic finger costed about 25 s. It is too long. For a human, this process may be 1-2s. Can the robot finger do the bending process in this time?*

Response: Actually, the robot finger can do the bending process as human. The revised data shown in Figure 3d. Related discussion is also added in the revised manuscript on page 7 line 12-13.

Q#8: *As the manuscript mentioned, the strength of the spider silk composite@10% SWCNT was about 1.5 GP which was better than normal steel fiber. But why the Loading weight of spider silk composite@10% SWCNT was lower than steel fiber in Figure 3h?*

Response: In our work, we used a bond of spider silk as the electro-tendon. Although the single spider silk has better mechanical properties than steel fiber, the bond of spider silk shows lower lifting weight than steel fiber. In addition, the lifting weight here we shown in the manuscript is the performance of robotic hand based on different fiber. The knots of the tendons and structure of hand have induced the lower light weight of S-silk than steel fiber.

Minor Issues #1: *In Figure 1g, the author should correct the unit (%) to (wt%).*

Response: The revised version is shown in Figure 1g.

Minor Issues #2: *In Figure 3h, there is a misunderstand. The author should correct the unit (kg) to (kg/mm²).*

Response: We have corrected the Figure 3h in our revised manuscript using the unit of N/mm².

REVIEWERS' COMMENTS:

Reviewer #1 (Remarks to the Author):

All comments from the reviewer were well addressed in the revised manuscript.

Reviewer #2 (Remarks to the Author):

Comments

The authors have well answered my questions and comments. It can be accepted now in my opinion.